

# Impact of particulate matter reductions on aerosol HO₂ uptake and rising surface ozone pollution in India

Gopalakrishna Pillai Gopikrishnan[1,2], Daniel M. Westervelt[2] and Jayanarayanan Kuttippurath[1]

[1]CORAL, Indian Institute of Technology Kharagpur, Kharagpur–721302, India.
[2]Lamont-Doherty Earth Observatory, Columbia University, New York - 10964, NY, USA

*Correspondence to:*
    Daniel M. Westervelt (*danielmw@ldeo.columbia.edu*)
    Jayanarayanan Kuttippurath (*jayan@coral.iitkgp.ac.in*)

**Abstract**. Atmospheric aerosols significantly contribute to air pollution and influence atmospheric chemistry, impacting air quality and public health. Decrease in aerosols can hinder the radical uptake sink of $HO_2$, and thus increase $NO_x$ and OH, and subsequently increase ozone levels. This study investigates the seasonal variations of $PM_{10}$ and aerosol surface area and their effect on surface ozone levels in India, using the GEOS-Chem Chemical Transport Model for the years 2018 and 2022, two years with high and low simulated $PM_{10}$ concentrations, respectively. The results reveal substantial seasonal variations in $PM_{10}$ and aerosol surface area. In winter (DJF), higher $PM_{10}$ and aerosol surface area in the Indo-Gangetic Plain (IGP) and western Central India (CI) result from biomass burning and industrial activity, while coastal regions show lower aerosol surface area. A decrease in aerosol surface area is seen during the pre-monsoon (MAM) and monsoon (JJAS), followed by an increase in the post-monsoon (ON) season. As a result, aerosol-induced $HO_2$ uptake during winter and post-monsoon lowers ozone concentrations by approximately 30 μg/m³ in 2022 when compared to that of 2018. In contrast, during monsoon in 2022, the decrease in aerosol surface area caused an ozone increase of 10–20 μg/m³ when compared to that of 2018. On average, eighty percent of this increase in surface ozone due to reduction in PM can be mitigated by reducing anthropogenic $NO_x$ emissions by 25–50%. Thus, we recommend integrated strategies addressing aerosols, precursor emissions and regional meteorology to combat ozone pollution.

## 1. Introduction

In recent decades, surface ozone ($O_3$) has gained significant research attention due to its role as a transient (ranging from a few hours to a few weeks) secondary pollutant and its detrimental impacts on human health and agricultural yield (e.g., Mills et al. 2007; Avnery et al. 2011; Rathore et al., 2023; Gopikrishnan and Kuttippurath, 2024). It is an important trace gas in tropospheric chemistry, facilitating oxidation processes as the principal source of hydroxyl radicals (OH) (Logan, 1985) while also involved in chemical reactions with various organic molecules (Anderson, 2007). Ozone in the troposphere is a secondary pollutant generated via photochemical reactions that involve carbon monoxide (CO) and volatile organic compounds (VOCs) together with nitrogen oxides ($NO_x$ = NO + $NO_2$), recognised as ozone precursors (Crutzen 1995; Gopikrishnan et al., 2022). However, these processes are dependent on solar radiation, with the reaction rate often peaking during summer months.



The formation dynamics of surface ozone is greatly affected by the presence of precursors such as $NO_x$ and VOCs, resulting in $NO_x$-limited and VOC-limited regimes (Fiore et al., 2002; Lu et al., 2019). In the $NO_x$-limited regime, the concentration of $NO_x$ is low compared to VOCs, implying that increasing VOC levels can promote further ozone generation due to inadequate $NO_x$ to completely react with VOCs. In a VOC-limited regime, the availability

of VOCs is restricted, and further VOC emissions will exert a negligible impact on ozone formation until $NO_x$ concentrations are high. Nevertheless, these regimes are significantly affected by local emission sources, atmospheric chemistry and climatic conditions. For instance, aerosols can influence these regimes by modifying the concentration of hydroxyl (OH) and hydroperoxyl ($HO_2$) radicals, affecting the equilibrium between $NO_x$ and VOCs and eventually determining the rate and extent of ozone generation in different locations (Jacob et al., 2005;

Feng et al., 2016; Wang et al., 2018; Westervelt et al., 2025).

Previous studies based on ground-based measurements from urban centers and industrial sites have shown substantial decrease in the particulate matter in the Indian cities post the implementation of National Clean Air Programme (NCAP), which primarily targets reducing the city-level PM pollution by 20–30% in 131 cities when compared to that of 2017–18. since 2019. Gopikrishnan and Kuttippurath (2024) studied the MDA-8 ozone

changes and observed that cities such as Visakhapatnam and Tirupati reported zero days of MDA-8 ozone surpassing 100 ppb post the implementation of NCAP in 2022 when compared to that of 2018 as the base year. Nevertheless, some cities in the Indo-Gangetic Plains (IGP) of India, for instance, Agra, Singrauli, Ghaziabad, with decreasing aerosol loading and particulate matter pollution (Gopikrishnan and Kuttippurath, 2025), show an increase in the surface ozone levels. This trade-off between particulate matter (PM) and surface ozone was also

mentioned in other highly populated regions of the world. For example, Wang et al. (2020) also observed that the opposite changes in surface ozone and $PM_{2.5}$ emerged as an unforeseen consequence of China's Clean Air Action Plan, aimed at reducing air pollution. Following the implementation of the plan, ozone levels increased over the summer in the North China Plain, due to reduced $NO_x$ emissions and steady or rising VOC emissions, alongside significant decreases in $PM_{2.5}$ concentrations. This indicates that addressing ozone pollution requires deeper

knowledge than broadly categorising it as $NO_x$-limited or VOC-limited, as the influence of PM and aerosols is significant (Zhao et al., 2023). Therefore, future air quality management systems worldwide must include the intricate relationship between ozone, particulate matter, and precursor emissions to effectively reduce both pollutants simultaneously.

Rural pollution is also a critical, yet often overlooked aspect of air quality management, especially in a country

like India, where a significant portion of the population resides in rural areas (Pathak and Kuttippurath, 2022; 2024). Rural regions are not immune to air pollution; they face unique challenges such as solid fuel combustion for cooking, agricultural residue burning and increasing vehicular emissions due to expanding road networks (Bhuvaneshwari et al., 2019; Chanana et al., 2023). These activities contribute to elevated levels of PM and ozone precursors, which can drift into urban areas, exacerbating pollution in cities through regional transport

mechanisms. Furthermore, rural air pollution has profound implications for public health, as rural populations often lack access to healthcare infrastructure to address respiratory and cardiovascular diseases caused by poor air quality (Coker and Kizito, 2018; Manisalidis et al., 2020). It also impacts agricultural productivity, with rising ozone levels reducing crop yields, thereby threatening food security in agrarian economies like India (Pandya et



al., 2022; Anagha et al., 2023). Addressing rural pollution is essential for achieving holistic improvements in air
quality and ensuring equitable environmental health benefits across urban and rural populations.

Studies on the impact of aerosols on near-surface ozone formation have predominantly focused on the significance
of Aerosol Optical Depth (AOD) extinction in photochemical processes that generate ozone (Bian et al., 2007;
Kim et al., 2013; Wang et al., 2019). AOD quantifies the degree to which aerosols scatter and absorb sunlight,
hence influencing the quantity of solar radiation that reaches the Earth's surface. This subsequently affects the
photochemical mechanisms that facilitate ozone synthesis in the lower atmosphere. Nevertheless, in areas with
high aerosol concentrations, the reduction of surface ozone is not only governed by AOD extinction, but also the
aerosol surface area by offering reactive sites for chemical reactions with molecules such as $HO_2$ (Macintyre and
Evans, 2011; Song et al., 2021). The interaction between aerosols and these species leads to the elimination of
ozone precursors, including OH radicals, which are vital for ozone production.

In this study, we aim to evaluate the impact of $PM_{10}$ and aerosol surface area on surface ozone pollution in India.
Since several studies are available on the air pollution concentrated in the urban regions of India, our analysis
therefore explores the ozone formation regimes across entire India, using a high-resolution GEOS-Chem Chemical
Transport Model. This allows us to examine the impact of regional and larger scale dynamics of ozone production,
which are often influenced by agricultural practices, biomass burning and long-range pollutant transport. By
simulating two distinct years—2018 (a high $PM_{10}$ year) and 2022 (a low $PM_{10}$ year)—and incorporating/excluding
the reactive absorption of $HO_2$ onto aerosols (via the $\gamma HO_2$ parameter), we aim to understand how reductions in
$PM_{10}$ levels influence aerosol surface area and the secondary ozone chemistry. Furthermore, we categorize ozone
generation regimes into $NO_x$-limited, VOC-limited and Aerosol Inhibited Regime (AIR) regions based on the
dominant termination reactions in surface ozone formation for each model grid. Additionally, by scaling down
$NO_x$ emissions by 25 and 50%, we assess the efficacy of national and local emission control measures in mitigating
$NO_x$ emissions and their subsequent impact on surface ozone levels. This approach provides important insights
into the chemistry between PM, precursor emissions and surface ozone across diverse geographical regions in
India, highlighting the need for integrated air quality management strategies beyond urban boundaries.

## 2. Data and Methods

### 2.1 Region of Study

India can be delineated into six regions based on topographical characteristics, demographic factors, and pollution
levels: Peninsular India (PI), Northwest (NW), Northeast (NE), Central India (CI), Indo-Gangetic Plain (IGP),
and Hilly Regions (HR), as shown in **Figure 1**. Peninsular India, which encompasses Kerala, Tamil Nadu,
Karnataka, Goa, Andhra Pradesh, and Telangana, has three of the ten most populous states—Andhra Pradesh,
Tamil Nadu, and Karnataka—and features substantial forest cover in the Western Ghats. Sources of pollution in
this region are automobile emissions, industrial operations and biomass combustion, especially in urban areas like
Bengaluru and Chennai. Central India, comprising Maharashtra, Madhya Pradesh, Chhattisgarh, Jharkhand and
Odisha, is marked by low population density yet is linked to thermal power plants, coal mines and steel factories.
Cities like Mumbai and Pune have higher pollution levels attributable to industrialisation and trash incineration.
Northwest India includes Rajasthan and Gujarat, characterised by dry landscapes like the Thar Desert; pollution



is attributed to dust from unpaved roads and construction activity, along with industrial emissions from sectors such as textiles. Northeast India comprises Tripura, Mizoram, Manipur, Nagaland, Meghalaya and Assam; despite its low population density and extensive vegetation, this region is affected by automobile pollution and slash-and-burn agriculture methods. The Indo-Gangetic Plain encompasses West Bengal, Bihar, Uttar Pradesh, Haryana and

Punjab; it is characterised by a high population density exceeding 1000 individuals per square kilometre and has substantial pollution resulting from heavy traffic, industrial effluents and extensive crop residue burning during harvest periods. The Hilly Regions encompass Jammu and Kashmir, Himachal Pradesh, Uttarakhand, Sikkim and Arunachal Pradesh. These areas are less industrialised and ecologically sensitive, with economies dependent on agriculture and forestry. They experience localised pollution from vehicular emissions and biomass combustion,

which is intensified by winter temperature inversions that confine pollutants near the surface (Kuttippurath et al., 2020; Gopikrishnan et al., 2022).

**2.2 GEOS-Chem**

GEOS-Chem is an extensive model developed for modelling complex oxidant-aerosol chemistry in the troposphere and stratosphere (Zhang et al., 2011; Kim et al., 2015). Employing the Kinetic PreProcessor (KPP)

3.0 as its chemical solver, GEOS-Chem integrates sophisticated functionalities via the FlexChem interface, facilitating a flexible methodology for chemical kinetics (Bates et al., 2023). The model complies with the most recent JPL/IUPAC guidelines for chemical mechanisms, with substantial modifications that improve the depiction of diverse chemical processes, including those related to isoprene, aromatics and nitrates (Bates et al., 2024). Recent advancements have enhanced the treatment of complex chemical reactions, including methanol synthesis

and mercury redox chemistry, facilitating more precise atmospheric forecasts. Additionally, the model incorporates the reactive absorption of $NO_x$ by aerosols and computes aerosol hygroscopicity, which is essential for comprehending aerosol-cloud interactions. GEOS-Chem offers a comprehensive framework for investigating atmospheric chemistry and its effects on air quality and climate change, accommodating various chemical species and reactions, including halogens and hydroxymethanesulfonate. This model provides thorough understanding

into ozone generation mechanisms and pollutant interactions, serving as an essential resource for addressing air quality challenges and comprehending the wider implications of atmospheric dynamics.

GEOS-Chem has been rigorously validated by researchers globally. For instance, Travis and Jacob, (2016) showed that, despite the successful simulation of ozone and its precursors in the SEAC4RS aircraft data below 1 km of altitude, MDA8 surface ozone was biased high in the model by +6 μg/m³ on average. David et al. (2019)

used the GEOS-Chem transport model in India and observed that the model reasonably simulated the tropospheric $O_3$ abundances and vertical profiles, with a mean bias of 1–3 DU compared to observations for the period 2000–2015. Christiansen et al. (2022) found that the GEOS-Chem model, when validated against ozonesonde, aircraft, and satellite observations globally, showed reasonable agreement with a mean bias of 1–3 DU for the period from 1990 to 2017. Mao et al. (2024) reported that the GEOS-Chem model, validated against MDA8 $O_3$ concentration

observations in central and eastern China from May to July 2017, showed a strong correlation of 0.77 (95% confidence level). In Nanjing, the simulated MDA8 $O_3$ concentrations converged with the observed trend, with a correlation coefficient of 0.65, a normalised mean bias (NMB) of 5%, and a normalised mean error (NME) of 21%. Karambelas et al. (2022) used the GEOS-Chem model to simulate $PM_{2.5}$ concentrations during high PM





episodes in the fall of 2015 and 2017, finding that the model underestimated observed concentrations by 28%,

with an average observed $PM_{2.5}$ concentration of 142 μg/m³ compared to 125 μg/m³ from the model. Despite this underestimation, the model demonstrated strong spatial correlations with observed concentrations ($r^2 = 0.67$), accurately capturing the spatial distribution of $PM_{2.5}$ across India. David and Ravishankara (2019) used the GEOS-Chem model to estimate boundary layer ozone ($BLO_3$) contributions across eight regions of the Indian subcontinent, finding that $BLO_3$ in northern India, Pakistan, and Sri Lanka is largely (about 65–70%) influenced

by regions outside the subcontinent. They also highlighted the growing importance of central India in contributing to ozone pollution, with regional meteorology playing a key role in the redistribution of $BLO_3$ and its precursors including Chlorine (Cl).

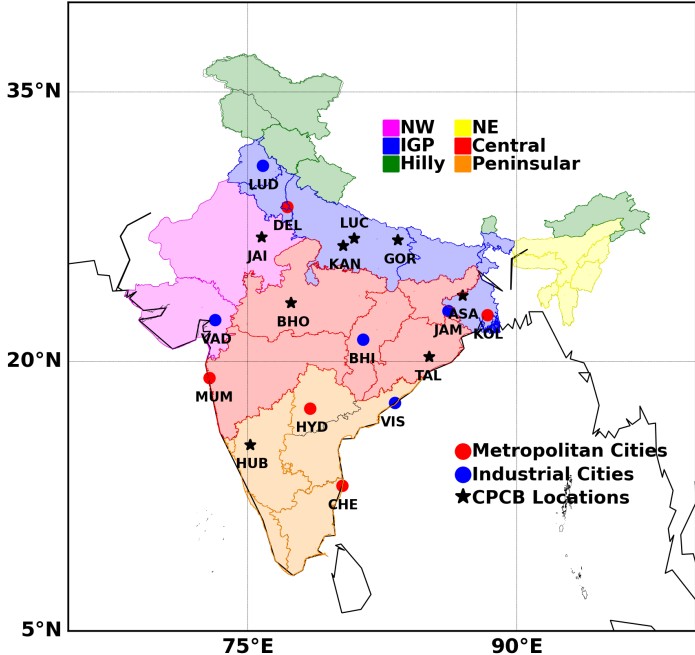

**Figure 1: The region of study. The regions considered are also shown in the respective colours. Here, NW is Northwest,**
**NEI is Northeast India and IGP is Indo Gangetic Plains. The star marks indicate the respective ground-based stations.**
**The red and blue dot indicates the metropolitan and Industrial cities in India. The letters represent the first three**
**letters of each city considered in the study (e.g. DEL is Delhi), and their expanded names are given in Figure S1.**

In this study, GEOS-Chem is simulated in a 2.0°x2.5° latitude by longitude grid resolution globally for the boundary conditions for the years 2018 and 2022. One year spin up simulations are used to generate the model

restart files and the boundary conditions. The nested model has a grid resolution of 0.25°x0.3125° latitude by longitude and a spin up period of 1 month. We use the data from the lowest level of the ETA coordinate system from GEOS-Chem model simulations to analyse the correlation between modelled and observed $PM_{10}$ and ozone data. We use ground-based measurements from eight monitoring sites situated in different regions of India, as shown in Figure 1, reflecting a wide array of meteorological conditions. The altitude of the lowest model level



may not consistently align with the actual altitude of ground-based monitoring locations, thereby causing inconsistencies between modelled and observed data. Therefore, we calculate the correlation coefficient for the time series of ozone and $PM_{10}$ values at each station to evaluate the strength of the association between these variables, as shown in Figure S1. The correlation coefficient between modelled and observed values of $PM_{10}$ and ozone is above 0.6 for most of the stations, except in Jaipur in the northwest, where the correlations are 0.5 and

0.4 for $PM_{10}$ and ozone, respectively. Although these moderate to strong correlations suggest a satisfactory alignment between the model and the facts, certain systematic differences remain. The GEOS-Chem model generally underestimates $PM_{10}$ concentration, possibly attributable to its 0.25°x0.3125° resolution, which fails to account for fine-scale local variations, including point sources of pollution. In contrast, the model often overestimates surface ozone concentrations for most of the stations considered here. Despite the model's higher

bias caused by uncertainties in emissions of anthropogenic and biogenic precursors, uncertainties in ozone sinks such as dry deposition, and the coarse resolution of the model limits its ability to capture ozone titration and other localised chemical processes (Westervelt et al., 2019). Our analysis is focused on the differences between the simulations for 2018 and 2022.

**2.3 Methods**

The aerosol uptake coefficient, which quantifies the efficacy of particles in absorbing reactive gases, substantially impacts surface ozone concentrations (Jacob, 2000). In areas with elevated aerosol concentrations, aerosols possessing a high uptake coefficient can more efficiently eliminate ozone precursors, including $HO_2$ and other reactive radicals, from the atmosphere (Li et al., 2018). These aerosols interact with atmospheric chemical species, reducing the levels of OH and $HO_2$ radicals, which are essential for ozone synthesis. The aerosol uptake coefficient

of $HO_2$ ($\gamma HO_2$), however, is affected by several factors, including the physical and chemical characteristics of the aerosol particles, such as size, composition and moisture content (George and Abbatt, 2010). These parameters show the efficacy of aerosol interactions with atmospheric radicals, thereby affecting ozone production.

Termination rates of chain reactions were determined using archived species concentrations and physical parameters such as temperature, pressure and humidity, in addition to aerosol properties, as in Ivatt et al. (2022).

Although we do not classify radical product generation in peroxyl-radical self-reactions as termination stages, we regard non-radical products as ongoing termination processes. The heterogeneous loss rate of $HO_2$ was assessed by evaluating the radius and surface area of different aerosol forms from the archived model outputs. For our simulations, we employed a default $HO_2$ reactive uptake coefficient ($\gamma HO_2$) of 0.2. Laboratory studies of pure synthetic aerosols indicate lower uptake coefficients ($\gamma HO_2 < 0.2$), whereas real-world aerosol studies reveal

values between 0.08 to 0.40, implying that elements such as transition metals may increase aerosol uptake (Kolb et al., 2010; Christian et al., 2018). We have also simulated the model with $\gamma HO_2$ to be zero to evaluate its impact on ozone formation mechanism. While we presumed $H_2O$ to be the exclusive product of $HO_2$ absorption, our results remain valid when $H_2O_2$ is considered. Employing a singular $\gamma HO_2$ value likely oversimplifies the variability, as existing models fail to account for its temporal oscillations (Stavrakou et al., 2013; Sheehy et al.,

205    2010).

The heterogeneous radical loss rate of $HO_2$ is then determined by iterating through the radius and surface area of various aerosol types, including organic carbon, black carbon, dust, sea salt, nitrate–sulfate–ammonium and



secondary organic aerosol (Ivatt et al., 2022). These factors, along with temperature and air density, were used to calculate the first-order loss rate of HO₂. Furthermore, emissions and photochemical reactions are strongly

affected by seasonal variations in India. The seasons are defined as Winter (December-January-February; DJF), Pre-monsoon (March-April-May; MAM), Monsoon (June-July August-September; JJAS) and Post-monsoon (October-November; ON).

**3. Results**

**3.1 Distribution of Surface Ozone and PM for the years 2018 and 2022**

**3.1.1 Annual Distribution**

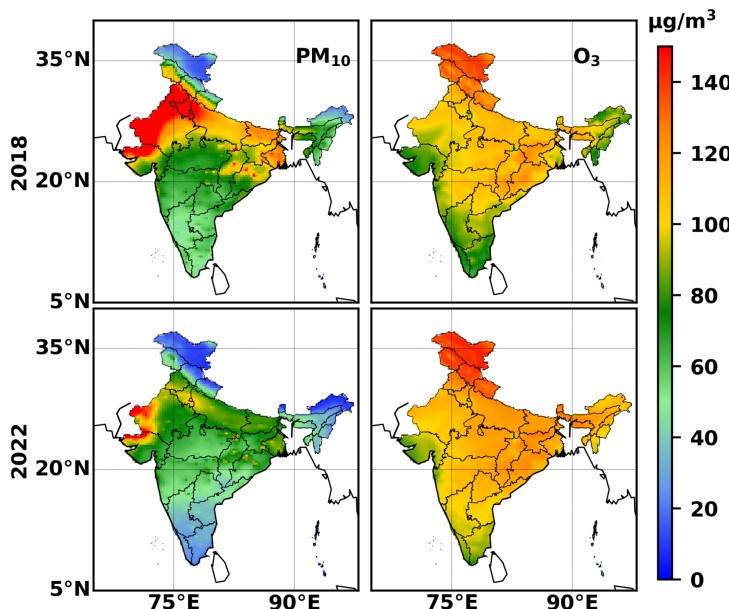

**Figure 2: Annual average distribution of PM₁₀ and ozone for the years (Top) 2018 and (Bottom) 2022.**

Figure 2 shows the annual average distribution of PM₁₀ and ozone in India, as modelled by GEOS-Chem at near-surface levels for the years 2018 and 2022. The peak PM₁₀ concentrations (130–150 μg/m³) are observed in the

NW dry regions, primarily owing to dust mobilisation from the Thar Desert (Santra et al., 2018; Tomar and Singh, 2023). This is consistent with previous studies indicating that dry and semi-arid areas are substantial suppliers of mineral dust, capable of being carried across extensive distances (Maji and Sonwani, 2022). IGP also has elevated PM₁₀ concentrations (120–140 μg/m³), but the levels are lower than those in the NW. This higher PM₁₀ in IGP can be attributed to extensive human activities, such as industrial emissions, vehicle exhaust and agricultural

emissions including biomass and crop-residue burning, which emit large amounts of PM and precursor gases (Devi et al., 2020; Hassan et al., 2023). The reduced PM₁₀ levels in the IGP relative to the NW indicate that, despite higher anthropogenic emissions, the dominant climatic conditions, including increased humidity and



precipitation, facilitate wet deposition and particle scavenging (Sen et al., 2017; Singh et al., 2023). CI has moderate $PM_{10}$ concentrations (60–100 µg/m³), however localised hotspots (130–150 µg/m³) are observed in

industrial regions such as Nagpur, highlighting the substantial impact of industry emissions on local air quality (Lokhande and Khan, 2021; Saini et al., 2023). The NE region shows lower $PM_{10}$ concentrations (40–70 µg/m³), presumably because of less population density, decreased industrial activity and higher vegetation, which serves as a sink for $PM_{10}$ (Guttikunda and Nishadh, 2022; Kumari et al., 2025). PI shows the lowest $PM_{10}$ concentrations (20–40 µg/m³), attributable to its coastline environment and the consequent effective ventilation from sea breezes.

The proximity to the Indian Ocean promotes the distribution and elimination of pollutants through improved air mixing and deposition mechanisms (Lavanyaa et al., 2023; Nilaya et al., 2024).

Surface ozone levels in India show considerable regional changes for the years 2018 and 2022. The peak surface ozone concentrations (120–140 µg/m³) are noted in HR. This can be attributed to the stratospheric-to-troposphere transport (STT) of ozone, when ozone-laden air is transported from the stratosphere into the troposphere. Previous

studies also show that STT occurrences occur with greater frequency and intensity in mountainous areas because of orographic lifting and tropopause folding (Phanikumar et al., 2017). Additionally, the reversal of winds in late autumn and early winter promotes the movement of ozone-laden air masses from stratospheric-tropospheric exchange (STE) zones in the Himalayan region, hence increasing surface ozone levels (Wang et al., 2024). Subsequent to the HR, higher ozone concentrations (100–120 µg/m³) are found in IGP and eastern CI. In IGP, the

secondary photochemical generation of ozone from anthropogenic precursors is the primary factor. IGP is marked by high levels of $NO_x$ and VOCs released from concentrated industrial zones, heavy automobile traffic and widespread agricultural practices, such as biomass and crop residue combustion (Payra et al., 2022; Sinha and Sinha; 2019). The high ozone levels in eastern CI are presumably affected by biogenic VOC emissions from its extensive vegetation, together with high anthropogenic pollution (Mahilang et al., 2021; Kuttippurath et al., 2022).

Stagnant meteorological conditions, including temperature inversions and minimal wind velocities, confine pollutants at the surface, facilitating ozone production by photochemical reactions in strong sunlight (Gopikrishnan and Kuttippurath, 2024). PI has the lowest surface ozone concentrations (80–100 µg/m³), due to robust ventilation and mixing owing to its proximity to the Indian Ocean. The maritime impact facilitates the redistribution of ozone and its precursors, nevertheless, high humidity, frequent precipitation and lower precursor

emissions constrain photochemical ozone formation (Li et al., 2019; Lakshmi et al., 2024). Furthermore, the degradation of ozone through interactions with water vapour and its deposition onto the ocean surface also results in lower ozone concentrations in coastal regions (Galbally and Roy, 1980).

### 3.1.2 Seasonal Distribution

Figure S2 and Figure S3 show the seasonal distribution of PM and ozone in India for the years 2018 and 2022.

$PM_{10}$ concentrations are considerably higher throughout IGP during the winter and post-monsoon seasons (DJF and ON, respectively), with values ranging from 100 to 140 µg/m³. This occurs mainly due to stable atmospheric conditions marked by low wind speeds and temperature inversions, which trap pollutants at the surface, inhibiting their dispersal (Mhawish et al., 2020; Paulot et al., 2022; Jayachandran and Rao, 2024). The increasing dependence on biomass combustion for heating, together with agricultural methods like the incineration of crop

residues post-harvest season, substantially contributes to the accumulation of $PM_{10}$ in the region at this time of



year (Payra et al., 2022; Mogno et al., 2021). Conversely, the NW region of India has its peak $PM_{10}$ concentrations during monsoon season. This is mostly because of the impact of dust storms originating from the Thar Desert and adjacent arid regions, which transport substantial amounts of mineral dust and $PM_{10}$ into the atmosphere (Yadav et al., 2022). Dust storms, frequently propelled by intense winds and high temperatures, can traverse large

distances, affecting air quality in both metropolitan and rural regions (Yu et al., 2023). In the pre-monsoon season, $PM_{10}$ levels are often higher throughout most of India, ranging from 60 to 100 μg/m³, with the NW region showing the highest concentrations, which is about 140–160 μg/m³. This results from a combination of factors, including local emissions from vehicle traffic, industrial operations and construction, alongside the persistent impact of dust storms and the long-range movement of pollutants from adjacent areas (Patil et al., 2013; Garg and Gupta, 2020).

The increased $PM_{10}$ levels during this period inflict considerable health hazards, especially for at-risk groups including children, the elderly, and those with pre-existing respiratory ailments (Adhikary et al., 2024; Pathak et al., 2024; Gopikrishnan and Kuttippurath, 2025).

Conversely, ozone levels also show a distinct seasonal pattern, peaking during the pre-monsoon, especially in CI, IGP and NEI, where it exceeds 140–160 μg/m³. This is mainly because of the high solar radiation during this

period, which catalyses photochemical reactions between $NO_x$ and VOCs, resulting in more ozone generation (Kutal et al., 2022). Precursor pollutants are released from diverse sources, including industrial complexes, automobile emissions, agricultural practices and biogenic origins such as vegetation (Sinha et al., 2014). Nevertheless, the coastal areas of India show reduced ozone levels during pre-monsoon, with concentrations between 80 and 100 μg/m³. In winter months, CI, especially eastern CI, has the highest ozone concentrations

(100–120 μg/m³), followed by IGP and NEI (80–100 μg/m³). This can be attributed to different factors including stable atmospheric conditions that confine pollutants near the surface and the impact of regional emissions from industrial and agricultural sources (David and Nair, 2011; Gao et al., 2020). The lowest ozone levels in India are recorded during the monsoon season. The western coast shows the lowest concentrations (60–80 μg/m³), followed by the NW (80–100 μg/m³), and NEI and IGP (120–140 μg/m³). The expanded cloud cover and precipitation

during monsoon season, as shown in Figure S4 and S5, reduce solar radiation, hindering photochemical ozone synthesis, while simultaneously facilitating ozone removal via wet deposition (Lal et al., 2014). Additionally, changes in wind patterns and atmospheric circulation during the monsoon season might facilitate the movement of cleaner air masses from the coastal environment, thereby further decreasing ozone levels in the subcontinent (Lu et al., 2018). In the post-monsoon season, ozone distribution is rather consistent in CI, NW and IGP, with

concentrations varying from 100 to 120 μg/m³. However, PI and NEI have slightly lower ozone concentrations (60–90 μg/m³) during this period. This period also sees a transition in weather patterns and a decrease in the factors that suppress ozone formation during the monsoon, leading to increase in ozone concentration due to weakening ozone upward transport associated with the summer monsoon retreat (Lu et al., 2018).

### 3.1.3 Comparison of $PM_{10}$ and Ozone levels

The relationship between $PM_{10}$ and ozone is inverse, attributable to their distinct formation mechanisms and atmospheric interactions (Jia et al., 2017). Higher PM levels, frequent during winter in areas such as IGP, often correlate with lower ozone concentrations (Song et al., 2022). PM disperses and absorbs sunlight, reducing the solar radiation essential for photochemical ozone formation, which depends on $NO_x$ and VOCs concentration in



the presence of intense sunlight. Black carbon (BC), a constituent of PM from combustion, can directly scavenge
ozone by acting as a deposition surface (Gao et al., 2018). Furthermore, stable atmospheric conditions including
temperature inversions that confine PM at the surface also impede vertical mixing, limiting ozone transport from
the upper atmosphere, hence facilitating the accumulation of ozone near the surface (Gopikrishnan and
Kuttippurath, 2025). Conversely, cleaner environments with reduced PM concentrations enhance ozone
generation due to the lack of light scattering and scavenging, which fosters photochemical activity (Rathore et al.,
2023; Sicard et al., 2023).

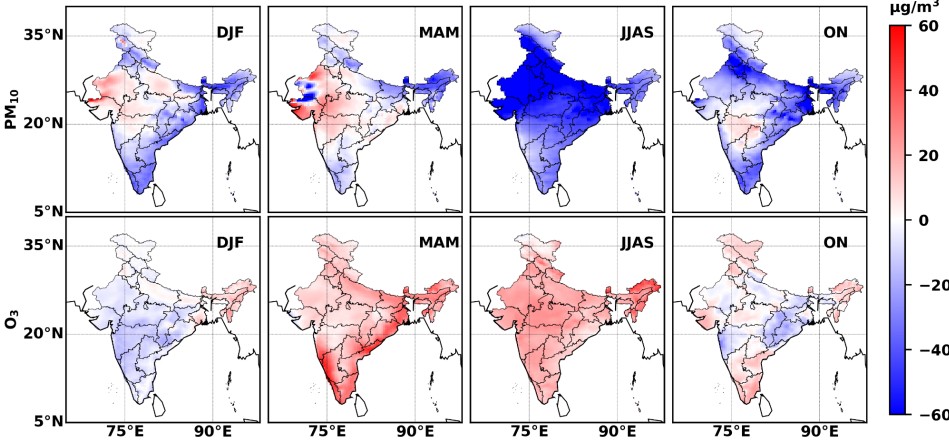

**Figure 3: The difference in (Top) PM$_{10}$ and (Bottom) Ozone in 2022 when compared to that of 2018.**

Figure 3 shows the change in PM$_{10}$ and ozone in India for the year 2022 when compared to that of 2018. In the
monsoon and post-monsoon seasons, PM concentrations drop by 40–60 μg/m³ in the NW, IGP and eastern CI,
while reductions of 10–30 μg/m³ occur in other regions, due to changes in the spatial distribution of precipitation
as shown in Figures S4 and S5 (Singh et al., 2023). The reduction in PM facilitates increased solar radiation
penetration, hence promoting photochemical ozone production, resulting in a substantial rise in ozone
concentrations (20–30 μg/m³ in most of India during monsoon and 20–30 μg/m³ in PI and western CI during post-
monsoon. The decrease in PM$_{10}$ reduces the surface area accessible for heterogeneous processes that might
otherwise reduce ozone levels (Gopikrishnan et al., 2025). Conversely, in pre-monsoon and winter seasons, PM
concentrations increase by 10–30 μg/m³ regionally, particularly in NW and central IGP, inhibiting ozone
formation due to reduced solar radiation and enhanced heterogeneous chemistry that depletes reactive species
such as HO$_2$ (Wang et al., 2024). The stagnant meteorological conditions associated with winter, including
temperature inversions, elevated humidity, and minimal wind speeds, intensify PM buildup, particularly in IGP,
resulting in further reductions in ozone levels (Gopikrishnan and Kuttippurath, 2024). Nonetheless, geographical
disparities remain, exemplified by a 20 μg/m³ reduction in PM in the Upper Northeast of IGP during pre-monsoon,
which causes a 20–30 μg/m³ rise in ozone levels, and a 10–20 μg/m³ decline in PM near the west coast, resulting
in a 40–60 μg/m³ increase in ozone. In winter, ozone variations are negligible (within 10 μg/m³) over much of
India; however, a reduction in PM (10–30 μg/m³) in the northeast correlates with a 20–30μg/m³ rise in ozone
levels. The seasonal variability is additionally affected by the aerosol-photolysis feedback, wherein PM influences



ozone by absorbing important species such as $H_xO_y$ and $NO_x$, hence reducing surface photolysis rates and inhibiting ozone production (Ivatt et al., 2022; Gopikrishnan et al., 2025).

## 4. Discussion

### 4.1 Changes in Aerosol surface area during 2018 and 2022

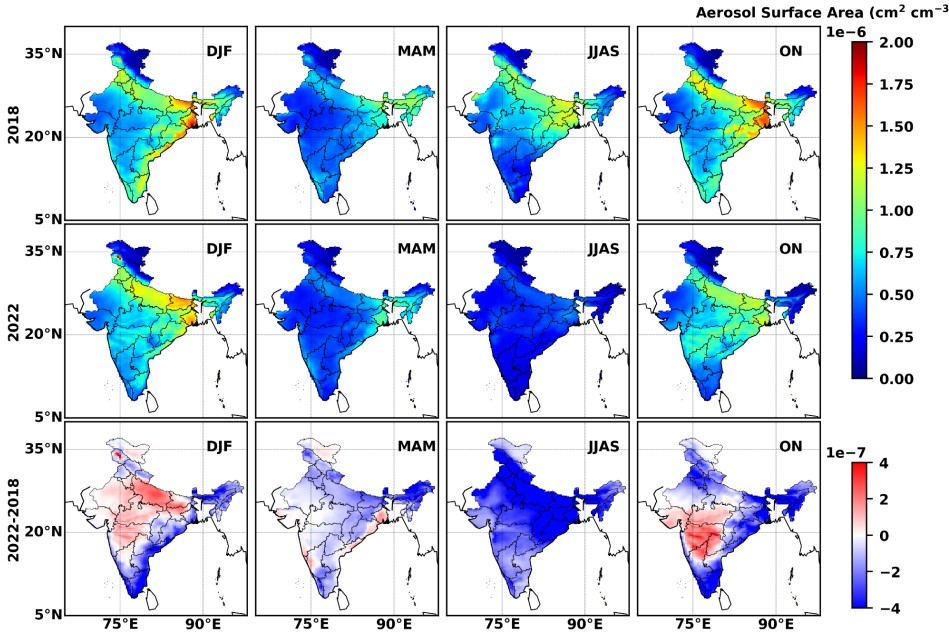


**Figure 4: Total Aerosol surface area in (Top) 2018 and (Middle) 2022. (Bottom) Difference between aerosol surface area for the year 2022 compared to that of 2018.**

Figure 4 shows the changes in distribution and the difference of aerosol surface area during the year 2018 and 2022. Figure S6 shows the difference in BC, OC, Sea Salt, sulphate and dust aerosols for the year 2022 when

compared to that of 2018. The aerosol surface area in India shows considerable seasonal variations driven by meteorological factors, human activities, and natural occurrences (Chen et al., 2021). In winter, IGP and western CI observe a notable rise of $2–3 \times 10^{-6}$ cm² cm⁻³ in aerosol surface area, mainly due to elevated levels of sulphate and organic carbon (OC). This is probably attributable to higher biomass combustion for residential heating, stagnant air conditions with reduced mixing heights that confine pollutants, and increased fossil fuel usage in

urban and industrial regions (Deshmukh et al., 2012). In contrast, coastal areas show a reduction of $0.5–1 \times 10^{-6}$ cm² cm⁻³, likely attributable to the influx of cleaner marine air masses, increased sea spray aerosol production affecting particle size distributions and intensified winds dispersing contaminants (Namdeo et al., 2024). The pre-monsoon season experiences a widespread reduction ($0–2 \times 10^{-6}$ cm² cm⁻³) in most Indian regions, because high solar radiation increases atmospheric mixing and photochemical reactions that facilitate aerosol removal

(Ramachandran and Cheriyan, 2008). Coastal regions observe a $1–2 \times 10^{-6}$ cm² cm⁻³ rise, primarily attributed to



increased sulphate concentrations, which can be due to high shipping emissions and industrial activity, together with enhanced biogenic aerosol precursors due to elevated temperatures (Athul et al., 2024). The reactions of sulphur dioxide ($SO_2$) with water vapour near the coastal regions will also lead to more sulphate aerosols near the coastal regions. western CI has elevated levels of mineral dust and sea salt aerosols, likely influenced by pre-monsoon dust storms from the Thar Desert, intensified convective activity elevating dust particles and enhanced sea-breeze circulation transporting marine aerosols inland (Ramachandran and Cheriyan, 2008). In monsoon season, a substantial decrease ($2-4 \times 10^{-6}$ cm² cm⁻³) in aerosol surface area is observed in several areas, especially in IGP, NE, NW and CI, with a slight reduction ($0-1 \times 10^{-6}$ cm² cm⁻³) observed in PI. This notable decrease can be attributed to effective wet deposition and scavenging mechanisms during monsoon rainfall, which remove aerosols from the atmosphere (Ramachandran and Cheriyan, 2008; Kedia and Ramachandran, 2011). Increased cloud cover also reduces the generation of photochemical aerosols (Sivaprasad and Babu, 2014). The post-monsoon season also shows a $2-4 \times 10^{-6}$ cm² cm⁻³ reduction in aerosol surface area in PI, eastern CI and the NE, as ongoing rainfall and vegetation regeneration reduce dust emissions. Western CI has an increase of $1-2 \times 10^{-6}$ cm² cm⁻³, with sulphate aerosols accounting for roughly 90% of this rise. It may be affected by post-harvest crop residue burning in agricultural zones following the monsoon, changing wind patterns dispersing pollutants from other locations, and a rise in industrial operations as economic activity resumes post-monsoon (Shaeb, 2019; Kajino et al., 2024). Nevertheless, variations in aerosol levels affect the surface area accessible for the heterogeneous absorption of $HO_2$ onto particles, hence impacting $HO_x$ radical concentrations and modifying the chemical processes that govern surface ozone generation (Westervelt et al., 2025). The efficacy of this absorption, affected by aerosol composition and surface area, can either inhibit or promote ozone production based on the $NO_x$ regime and aerosol loading (Xing et al., 2017).

**4.2 Impact of $HO_2$ uptake onto aerosol on surface ozone**

The GEOS-Chem model simulations for 2018 and 2022, performed with and without aerosol uptake of $HO_2$, provides key insights into the impact on surface ozone concentrations in India in the absence of this mechanism. The results show that the aerosol absorption of $HO_2$ substantially influences ozone concentrations, as shown in Figure 5, especially in areas like the IGP and eastern CI. The decrease in ozone concentrations in these regions, attributed to the absorption of $HO_2$ by aerosols, varies from 20 to 30 μg/m³, with the most significant changes during winter. This season is defined by increased aerosol surface area, which promotes heterogeneous processes that enhance the removal of $HO_2$, hence restricting its availability for ozone formation (Dyson et al., 2022). In 2022, aerosol concentrations rose substantially in winter when compared to 2018, especially in the middle IGP and western CI. This augmentation of aerosol surface area in 2022 inhibited ozone generation by about 5 μg/m³ in these areas. Subsequent to winter, the most notable changes in ozone concentrations were during post-monsoon, with decreases of about 10–20 μg/m³, especially in IGP and western CI. In 2022, there was a rise in aerosol surface area in western CI compared to 2018, resulting in an additional suppression of ozone generation by 5–10 μg/m³. The pre-monsoon season period had similar changes in ozone concentration, with decreases of about 5–10 μg/m³, whereas the monsoon period observed minimal variation, with ozone concentration changes between 0 and 5 μg/m³. Thus, the changes in ozone concentrations are directly influenced by aerosol surface area, which amplifies the heterogeneous elimination of $HO_2$, thereby restricting its availability for ozone synthesis (Anurose et al., 2024). Seasonal fluctuations in aerosol load, influenced by factors like monsoon dynamics, atmospheric transport,



and emissions, greatly influence the effects of aerosol-HO$_2$ interactions on ozone levels. The rise in aerosol
concentrations in 2022, especially during winter and post-monsoon, can be attributed to higher anthropogenic
activities increasing sulphate, BC and OC, which enhance the suppression of ozone production in these seasons
(Austin et al.,2015; Zhang et al, 2023).

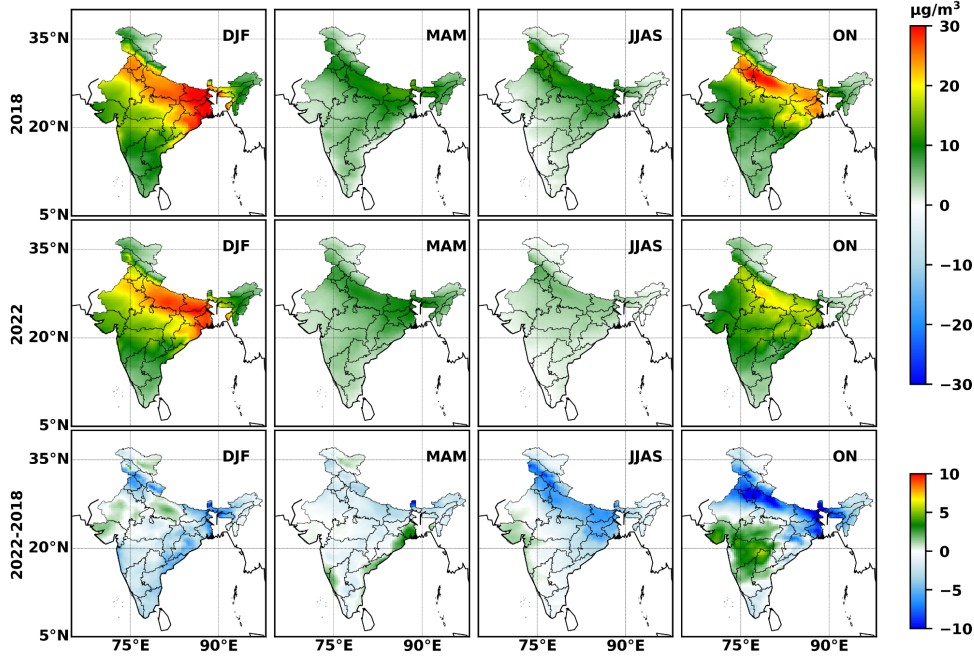

**Figure 5: Change in surface ozone with and without Aerosol uptake in (Top) 2018 and (Middle) 2022. (Bottom)
Difference between ozone contribution from aerosol uptake for the year 2022 when compared to that of 2018.**





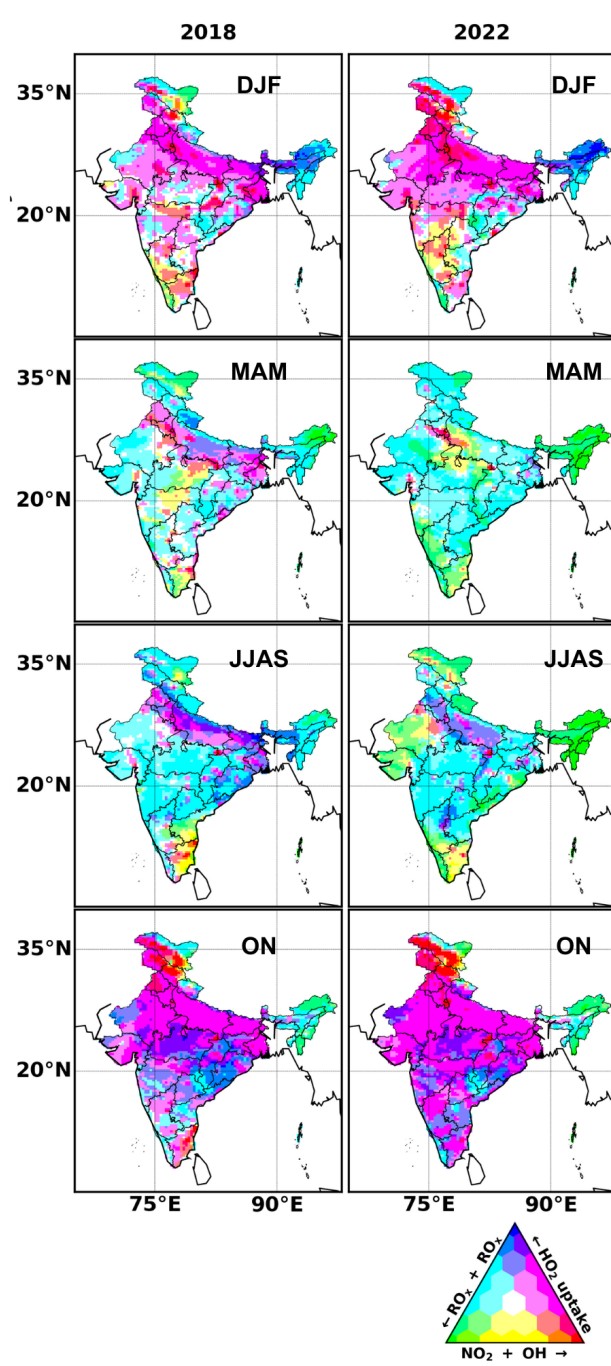

**Figure 6: Mean fraction of radical termination at the surface that occurs through OH + NO₂ (red), peroxyl-radical self-reactions (green) and aerosol uptake of HO₂ (blue) for the year (left) 2018 and (right) 2022.**




### 4.3 Surface Ozone Formation Regimes

Figure 6 shows the surface distribution of the relative chemical fluxes associated with the termination steps of ozone formation in India for the years 2018 and 2022. During monsoon in 2022, peroxyl-radical self-reactions (green) dominate most of PI, NE and NW arid regions, while in 2018, ozone termination was equally influenced

by $NO_2$+OH reactions and peroxyl-radical self-reactions in PI, whereas NW and NE by $HO_2$ uptake onto aerosols (blue). The dominance of peroxyl-radical self-reactions in urban areas such as Delhi, Bhiwadi, Vadodara, Jamshedpur and Nagpur indicates that $NO_x$ emissions in these cities limit ozone formation, promoting pathways where peroxy radicals neutralize ozone. This is typical of regions where high $NO_x$ levels suppress the production of ozone by inhibiting $HO_x$ chemistry (Romer et al., 2018). During winter, $NO_2$+OH termination and the uptake

of $HO_2$ on aerosols became prominent over much of India, particularly in IGP. The increased aerosol surface area during this period enhances the uptake of $HO_2$, thereby inhibiting ozone production. The presence of aerosols, often exacerbated by biomass burning and dust transport, amplifies this effect in areas where aerosols are abundant (Ivatt et al., 2022; Singh et al., 2018).

In post-monsoon, the termination of ozone formation shifts predominantly to $HO_2$ uptake across many regions,

particularly in IGP, CI, and parts of PI. The NEI, however, is an exception, where peroxyl-radical self-reactions continue to dominate the termination process. This variation in termination pathways can be attributed to regional differences in aerosol loading and the influence of seasonal emissions, including biomass burning during the post-monsoon season, which contributes to higher aerosol concentrations in IGP and CI. During the pre-monsoon period in 2022, the southern parts of northern IGP are predominantly influenced by $HO_2$ uptake, whereas the rest

of IGP and much of India find dominant termination by peroxyl-radical self-reactions. The increased aerosol surface area and the presence of fire events, particularly in the dry regions of NW and NE regions, enhance the uptake of $HO_2$, suppressing ozone formation (Kumari et al., 2025). Fire events and desert dust also contribute to higher aerosol concentrations, reinforcing the impact of heterogeneous reactions on ozone chemistry (Dyson et al., 2023; Dhanurkar et al., 2024). Overall, the termination mechanisms of ozone formation in India are strongly

influenced by regional aerosol levels, fire events and desert dust, which vary seasonally and modulate the balance between gas-phase and heterogeneous processes.

Furthermore, Figure 7 splits the domain based on the local largest termination step. In regions characterized as 'NO$_x$-limited,' the peroxy self-reactions dominate, making reductions in nitrogen oxides ($NO_x$) the most effective strategy for mitigating ozone pollution; this contrasts with 'VOC limited' regimes, where VOC reductions are

more beneficial, with the transition between these regimes dictated by the atmospheric concentrations of $NO_x$ and VOCs. Aerosol uptake of hydroperoxyl radicals ($HO_2$) on aerosol surfaces can lead to an 'aerosol inhibited' environment, diminishing ozone formation in areas with elevated aerosol concentrations.





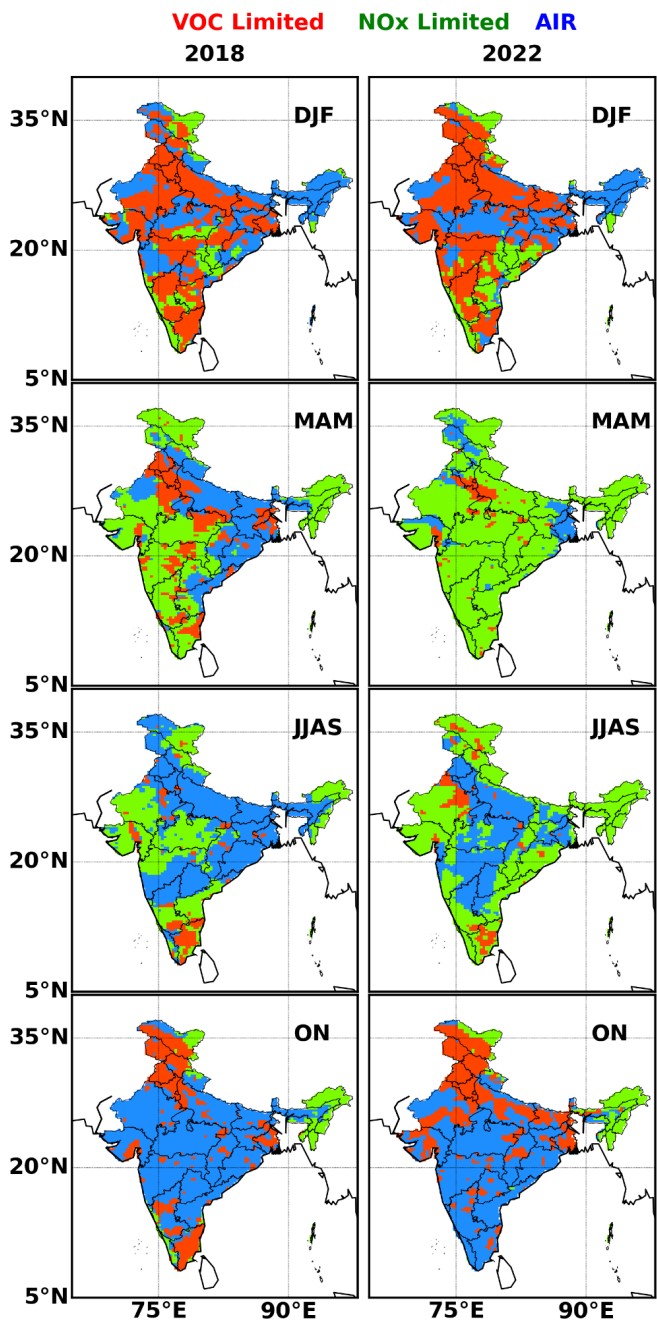

**Figure 7: The regional distribution of various ozone generation photochemical regimes modelled with γHO2 of 0.2 in**

**(left) 2018 and (right) 2022. Here, AIR is Aerosol Inhibited Regions.**




In 2018, much of IGP and CI were in an AIR, where the presence of aerosols, particularly in regions with high anthropogenic emissions and biomass burning, suppressed ozone formation by promoting the heterogeneous uptake of $HO_2$. However, a substantial reduction in aerosol surface area in 2022, shifted some of these regions into a $NO_x$-limited regime (for e.g. eastern CI, eastern IGP), where the availability of $NO_x$ becomes the limiting

factor for ozone formation. This transition to a $NO_x$-limited regime is associated with a subsequent increase in ozone concentrations in 2022 compared to 2018, as the reduction in aerosols resulted in more ozone to form via the photochemical pathways involving $NO_x$ and VOCs. The decrease in aerosol surface area, coupled with reduced aerosol-induced $HO_2$ removal, enhanced the ozone production efficiency, leading to higher ozone levels in these regions.

In contrast, during winter and post-monsoon, there is less to no difference in the dominant regimes between 2022 and 2018. In winter, most of India remains in a VOC-limited regime, where VOCs (such as those from biogenic sources and fossil fuel emissions) limit ozone formation. CI and NEI continue to be in AIR during this period, with high aerosol concentrations suppressing ozone production through $HO_2$ uptake. During post-monsoon, the majority of India remains in AIR, except for northern IGP, which is in a VOC-limited regime, and NEI, where the

region is predominantly $NO_x$-limited. The seasonal consistency in winter and post-monsoon suggests that the aerosol levels together with the overall seasonal and regional variations in emissions, meteorology and chemistry play a larger role in initiating the ozone formation regime in these periods. Therefore, while aerosol changes in 2022 had a notable impact on ozone production during monsoon, they had a lesser influence on the seasonal regimes in winter and post-monsoon.

**4.4 Impact of the national efforts in reducing $NO_x$ and its effect on surface ozone pollution**

National and local efforts to reduce $NO_x$ emissions in India have had a noticeable impact on surface ozone pollution, particularly in urban and industrialized regions (Chen et al., 2021; Misra et al., 2021; Gopikrishnan et al., 2022). $NO_x$, primarily emitted by vehicular traffic, power plants and industrial processes, plays a critical role in the formation of surface ozone through photochemical reactions. By reducing $NO_x$ emissions, several regions

in India have observed a significant decrease in ozone formation. National initiatives such as stricter emission standards for vehicles, the promotion of cleaner fuels, and the adoption of advanced technologies in industrial operations have contributed to the reduction in $NO_x$ emissions (UNESCAP, 2022). These efforts are particularly impactful in regions like IGP, which has been a hotspot for high $NO_x$ concentrations and consequent ozone pollution. Furthermore, policies aimed at reducing vehicular traffic, such as improved public transport systems

and the promotion of electric vehicles, have shown promise in lowering $NO_x$ levels and mitigating surface ozone formation (Saikia, 2025). These measures help disrupt the photochemical cycle that leads to high ozone levels, as lower $NO_x$ concentrations slow down ozone generation. However, the success of these efforts is often offset by increasing emissions from other sectors, such as agriculture, where practices like biomass burning continue to release $NO_x$ (Usmani et al., 2020). Despite these challenges, sustained efforts to reduce $NO_x$ emissions have

contributed to a gradual improvement in air quality, particularly in urban centers, leading to a secondary reduction in surface ozone pollution in time. Nevertheless, more stringent emission reductions are required to tackle the increasing ozone problem in India. Therefore, we scaled down the emissions of $NO_x$ in the model simulations to see how this decrease in $NO_x$ would facilitate the ozone generation or destruction in different regions of India.





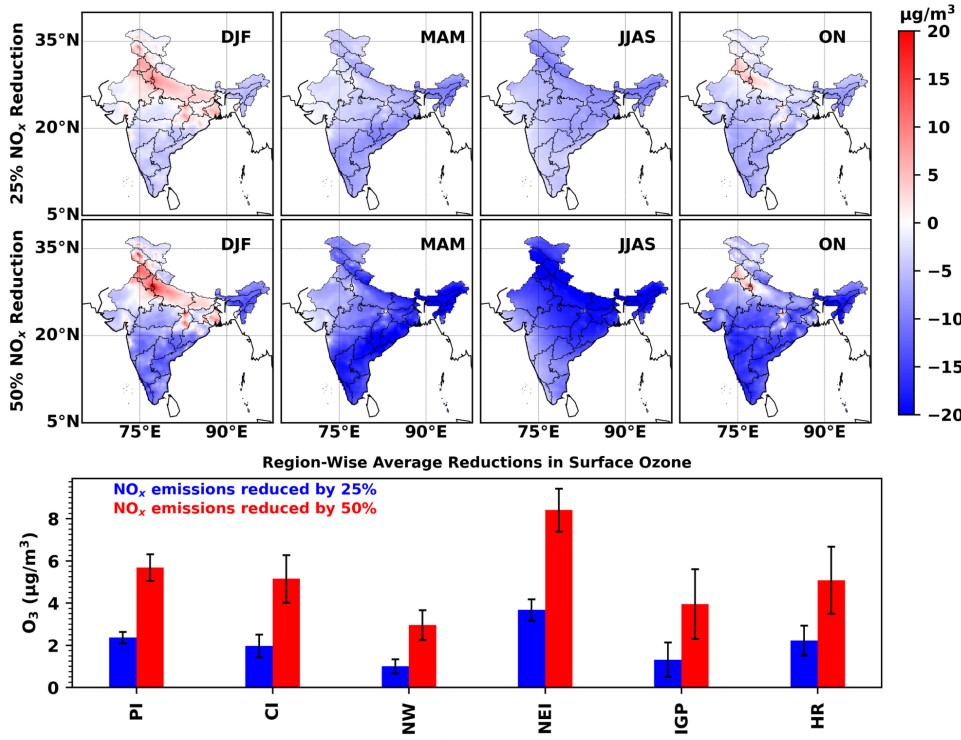

**Figure 8: The change in ozone levels when NOx emissions are scaled down by (top) 25% and (middle) 50% when compared to that of base simulation in 2022 with no reduction of emissions. (bottom) Region-wise averaged surface ozone reduction when NOx emissions are scaled down by 25 and 50% for the year 2022. The whiskers represent the standard deviations. Here, PI is Peninsular India, CI is Central India, NW is NorthWest, NEI is North East India, IGP is Indo-Gangetic Plains and HR is Hilly Regions.**

Figure 8 shows the difference in surface ozone in 2022 with 25% and 50% reduction in $NO_x$ emissions when compared to that of 2022 with no reduction in emissions. A decrease of approximately 5–10 µg/m³ in ozone levels is found with a 25% reduction in $NO_x$ emissions, while a 10–15 µg/m³ decrease occurs with a 50% reduction in $NO_x$ emissions in India. The most substantial reduction occurs during the monsoon season, characterised by a $NO_x$-limited regime prevalent throughout much of India, succeeded by the post-monsoon and pre-monsoon periods. During winter, particularly in IGP and eastern CI, ozone levels increase, as these regions are limited by VOCs during this period. The increase is about 5–10 µg/m³ with a 25% decrease in $NO_x$ emissions, and about 10–15 µg/m³ with a 50% reduction. Additionally, regional variations in ozone reduction are more pronounced in the 50% decrease scenario, while the 25% reduction shows a more uniform decrease across India (5–10 µg/m³). The reduction of ozone is most noticeable in PI during pre-monsoon and post monsoon (10–15 µg/m³), but, the ozone reduction occurs primarily during monsoon in IGP (10–15 µg/m³), with the smallest levels (5–10 µg/m³) in PI. The seasonal change can be due to decreased $NO_x$ levels in the PI during the monsoon, as rainfall significantly removes precursors in the atmosphere during this season. The findings indicate that roughly 75–80% of the ozone increase attributed to the decrease in PM and aerosols in India could possibly be alleviated through



enhanced regulation of NO$_x$ emissions. Also, the findings suggest that winter strategies must focus on reducing
VOCs, whereas stricter NO$_x$ control efforts are essential in other seasons to mitigate ozone pollution.

## 4. Conclusions

This study examines the interactions between PM$_{10}$, aerosols and surface ozone concentrations in India, focusing
on the changes modelled between 2018, a higher PM$_{10}$ year, and 2022, a lower PM$_{10}$ year. The findings highlight
that the seasonal and regional variability of PM$_{10}$ and aerosol surface area play a major role in modulating surface
ozone formation. During winter and post-monsoon seasons, elevated PM$_{10}$ and aerosols, particularly in IGP and
CI, led to higher uptake of HO$_2$, reducing its availability for ozone production and hence suppressing ozone levels
by about 30–40% when compared to the surface ozone levels in 2018 and 2022. However, during monsoon and
pre-monsoon seasons, reduced aerosol surface area and PM concentrations enhance photochemical ozone
formation, as the decrease in aerosol surface area allows more HO$_2$ to participate in ozone production. The shift
from an aerosol-inhibited regime to a NO$_x$-limited regime in 2022, due to lower aerosol pollution compared to
2018, resulted in an increase in ozone levels in several regions, especially in the IGP contributing to an increase
in its levels by about 20–30 µg/m$^3$. This suggests that the balance between aerosols, NO$_x$ and VOCs dictates ozone
formation, with aerosol concentrations continuing to significantly influence ozone levels during these seasons.
Nevertheless, local emissions from vehicular traffic, regional transportation, industrial operations, household
sources and agricultural residue combustion substantially control PM$_{10}$ and ozone levels, highlighting the
necessity for thorough, region-specific emission control measures. Furthermore, scaling down the NO$_x$ emissions
in the model simulations by 25 and 50% shows a reduction of about 5–10 and 10–15 µg/m$^3$ of surface ozone in
India, respectively. Thus, about 80% of the total increase in surface ozone due to the decrease in PM can be
reduced by additional efforts in reducing anthropogenic NO$_x$ emissions. Therefore, this study recommends the
need for integrated air quality management strategies that consider both aerosol and precursor emissions, along
with regional meteorological patterns, to address ozone pollution in India effectively.

*Data availability.*  GEOS-Chem model is available via https://geos-chem.readthedocs.io/en/stable/ and CPCB
data is available via https://app.cpcbccr.com/ccr/

*Authorship Contributions.* GSG: Writing – review & editing, Writing – original draft, Visualization, Validation,
Software, Methodology, Investigation, Formal analysis, Data curation. DMW: Writing – review & editing,
Writing – original draft, Supervision, Visualization, Validation, Software, Methodology, Investigation,
Conceptualisation. JK: Writing – review & editing, Writing – original draft, Supervision, Visualization,
Validation, Methodology, Investigation, Conceptualisation.

*Acknowledgements.* We thank all the data managers and the scientists who made available those data for this
study. GSG acknowledges the Prime Minister's Research Fellowship (PMRF; Grant No: 2402787), Ministry of
Education, India, for funding his Doctoral study at IIT KGP and the United States India Education Foundation for
his grant through the Fulbright-Kalam Climate fellowship for Doctoral Research (Grant No: 3067/FNDR/2024-
2025) at the Lamont-Doherty Earth Observatory and Columbia Climate School, Columbia University, New York.



JK and GSG thank the Director, Indian Institute of Technology Kharagpur (IIT Kgp), HoC, CORAL IIT Kgp and the Ministry of Education (MoE) for facilitating the study.

*Competing interests.* One author (JK) is an editor of ACP. The authors declare there is no other competing interest.

*Financial Support*. This project was supported by National Science Foundation grant OISE 2020677.

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
