# Peer review of "Impact of particulate matter reductions on aerosol HO2 uptake and rising surface ozone pollution in India"

_EGUsphere, 2025_

## Referee Comment (RC1)

**Major Comments**

In this manuscript, the authors use the GEOS-Chem model to look at the impacts of aerosols on surface ozone levels in India in two years, one with low and one with high PM10. They find that ozone decreases by ~30 μg m$^{-3}$ in the year with high aerosol pollution during the winter and post-monsoon. During the monsoon, the decrease in aerosol causes an increase in ozone of 10-20 μg m$^{-3}$. PM can be mitigated by reducing anthropogenic NO$_x$. This paper could be appropriate for publication in ACP with major revisions.

My main comments/concerns are as follows:
1. The authors go into too much detail that is not relevant to their specific study that makes the paper difficult to read. The authors should focus their discussion on the points they are trying to make. For example, they can shorten in particular Section 3 and just focus on when ozone and aerosols are high and impacts might be large. There is no need to get into a large overview of all aspects of India. Section 4.1 is not really needed until you discuss results. Use the information in Section 4.1 as needed to explain large impacts in Section 4.2.
2. There seems to be more data available in India than is used in this work. According to this paper, there are 123 sites with PM2.5 from www.openaq.org. https://pubs.acs.org/doi/epdf/10.1021/acsearthspacechem.2c00150?ref=article_openPDF.
3. This study would be more powerful if it better addressed trends. How much can reduced aerosols explain increased ozone or help improve model ability to capture trends? Overall, the paper needs at a minimum to show that the model can capture the relative difference in observed ozone and PM10 between 2018 and 2022.
4. The authors should be careful about directly relating changes in PM10 to changes in ozone. How much for example do differences in temperature and biogenic VOCs impact the ozone difference? Or meteorological factors such as windspeed and transport alter concentrations as well as PM10 levels? The authors should better describe their model setup, and it may be more appropriate to remove the effect of meteorology and focus on emissions by running 2018 and 2022 with the same meteorology.
5. Previous work has found that γHO2 of 0.2 might be too high in East Asia https://doi.org/10.5194/acp-23-2465-2023 and references within. How much would a value of 0.1 impact your results? Please include a discussion of the uncertainties surrounding this uptake coefficient. Also see this recent paper on the large variability in γHO2 depending on atmospheric conditions. https://pubs.rsc.org/en/content/articlehtml/2024/ea/d4ea00025k

**Specific Comments**

Page 2, line 38:

The authors state: "In the NO$_x$-limited regime, the concentration of NO$_x$ is low compared to VOCs, implying that increasing VOC levels can promote further ozone generation due to inadequate NO$_x$ to completely react with VOCs." This in incorrect. In the NOx-limited regime, the concentration of NOx is low relative to VOCs, meaning that ozone production is limited by the availability of NOx and increasing VOC levels has little effect. Similarly, the authors state "In a VOC-limited regime, the availability of VOCs is restricted, and further VOC emissions will exert a negligible impact on ozone formation until NO$_x$ concentrations are high." This is also incorrect. If ozone chemistry is VOC-limited, adding VOCs will increase ozone production.

Page 2 line 45 – The authors should cite https://www.nature.com/articles/s41561-022-00972-9 and https://www.pnas.org/doi/abs/10.1073/pnas.1812168116 where appropriate in this manuscript.

Page 2, line 49 – There must be a typo in 2017–18. since 2019.

Page 2, line 51 – How many exceedances were there in 2018?

Page 3, line 78 – What about the impact of aerosols on modifying PBL dynamics? I believe there is a large body of work also on this topic (e.g., https://academic.oup.com/nsr/article/4/6/810/4191281).

Page 3, line 94 – Cite https://www.nature.com/articles/s41561-022-00972-9 again here.

Page 4, line 124 – It is good practice to give the model version number and DOI here.

Page 4, line 125 – FlexChem is an option. Do you actually use it? If not don't cite it.

Page 4, line 130 - Cite the associated papers, although these advancements don't seem relevant to this work. Generally, just cite the relevant updates that impact your work.

Page 4, paragraph starting on line 138 – It is not necessary to cite GEOS-Chem papers from other regions. Just focus on India, and if necessary other parts of Asia. Other papers to include on India: https://pubs.acs.org/doi/full/10.1021/acsearthspacechem.2c00150. It is most important to show that your model setup has good performance against observations.

Page 5, paragraph starting on line 163 – Please provide information on what emissions you have chosen for India, and what scheme you are using to calculate aerosol (simple, complex, etc.). Also please provide information on the observations. Only PM10 is available? How did you calculate PM10 from GEOS-Chem individual species? It looks like there is ozone in the supplement as well? What is the instrument used, and the temporal resolution?

Figure S1 – Typically people show observations in black, and the model in red.

Page 6, line 180 – If the model is biased that high, there is likely something wrong with emissions. If you are not going to be able to investigate that, then it would be important to show that the model can capture the 2018 vs. 2022 difference to provide support for the relative impacts on ozone from aerosol differences.

Figure 2 – Could you put the observations as colored circles on top of the model? This would help with the previous comment. It is also important if you are going to discuss drivers of the pollution hotspots to show confidence in the model distribution.

Page 9, line 300 – Be careful of stating too much causation here. In winter there is also low sunlight, and cooler temperatures that promote nitrate formation.

Page 10, line 305 – BC scavenging of ozone is not in models, how important is it?

Page 10, line 307 – This is not an inverse relationship then if both PM and ozone accumulate near the surface. Also, you haven't mentioned the impact on the PBL height from aerosols (e.g., https://academic.oup.com/nsr/article/4/6/810/4191281). Also, if the PBL is suppressed, you might get ozone titration from high $NO_x$ emissions.

Page 11, line 343 – Does the model have different emissions in 2022 or 2018, or are the same emissions (e.g. for biomass combustion) used for both years?

Page 15, line 407 – This is opposite the conclusion of Ivatt et al., 2022, who found that Delhi is VOC-limited. Looking at your map, it looks like Delhi could be either a purple color or red color depending on the season, which would be VOC-limited or aerosol-inhibited.

Page 15, line 408 – The authors state that "The dominance of peroxyl-radical self-reactions in urban areas such as Delhi, Bhiwadi, Vadodara, Jamshedpur and Nagpur indicates that NOx emissions in these cities limit ozone formation, promoting pathways where peroxy radicals neutralize ozone. This is typical of regions where high NOx levels suppress the production of ozone by inhibiting HOx chemistry (Romer et al., 2018)." This is incorrect! High-NOx regions suppress ozone because the dominant loss is NO2 + OH, but this means that these cities are VOC limited.

On Figure 7 – It looks like Delhi is in the VOC-limited or aerosol-limited regime, depending on season. This contradicts the statement on Line 406 that Delhi (and other urban areas) are NOx-limited.

Page 17, line 442 – Again there is a need to be careful here as it is unlikely that 100% of the ozone difference is due to the reduction in aerosols.  One possible way to remove the impact of meteorology would be to use the same year of meteorology but with different emissions, assuming that different emissions are used in these two years.

Page 17, line 454 – Can you give more specific reasons for why the monsoon season would be more susceptible to aerosol uptake than winter or post-monsoon?

Paragraph starting on Page 18, line 480 – How much does reduced $NO_x$ also reduce aerosol?  What is the composition of aerosol?  Is there a lot of ammonium nitrate?

---

## Author Comment (AC1)

**RESPONSE TO REVIEWERS**

Thank you for your valuable time, effort and insightful comments. We have attended all suggestions and revised the MS accordingly. The author's response below and revised parts in MS are shown in blue typeface. We do hope that the referees and the Editor would find the revised MS more interesting and recommend a publication in *Atmospheric Chemistry and Physics* very soon.
* * *
**Response to Reviewer #1**

In this manuscript, the authors use the GEOS-Chem model to look at the impacts of aerosols on surface ozone levels in India in two years, one with low and one with high PM10. They find that ozone decreases by ~30 µg m$^{-3}$ in the year with high aerosol pollution during the winter and post-monsoon. During the monsoon, the decrease in aerosol causes an increase in ozone of 10-20 µg m$^{-3}$. PM can be mitigated by reducing anthropogenic NOx. This paper could be appropriate for publication in ACP with major revisions.

Thank You for the comments. Please see the specific response for each comments below:

**Major Comments:**

1. The authors go into too much detail that is not relevant to their specific study that makes the paper difficult to read. The authors should focus their discussion on the points they are trying to make. For example, they can shorten in particular Section 3 and just focus on when ozone and aerosols are high and impacts might be large. There is no need to get into a large overview of all aspects of India. Section 4.1 is not really needed until you discuss results. Use the information in Section 4.1 as needed to explain large impacts in Section 4.2.

Done. We have **shortened section 3 (Lines 267–351)** and moved the detailed discussions in **Section 3** to **section 1** in the supplementary data. Additionally, we have also moved **section 4.1** to **section 2** in the supplementary data.

2. There seems to be more data available in India than is used in this work. According to this paper, there are 123 sites with PM2.5 from www.openaq.org.

Done. As per the suggestion and data availability for these two years (2018 and 2022), **Figure 2** has been updated using data from a larger number of monitoring stations across India to allow for better spatial comparison. A discussion is also provided in **lines 225–230** and **section 3.1.1.**

3. This study would be more powerful if it better addressed trends. How much can reduced aerosols explain increased ozone or help improve model ability to capture trends? Overall, the paper needs at a minimum to show that the model can capture the relative difference in observed ozone and PM10 between 2018 and 2022.

Done. We have now added two new figures, **Figure S4** and **S5**, which compares the difference between the modelled and observed values of both ozone and PM$_{10}$ for the years 2018 and 2022. A corresponding discussion has been included in the revised MS, **lines 224–234**.

4. The authors should be careful about directly relating changes in PM10 to changes in ozone. How much for example do differences in temperature and biogenic VOCs impact the ozone difference? Or meteorological factors such as wind speed and transport alter concentrations as well as PM10 levels? The authors should better describe their model setup, and it may be more appropriate to remove the effect of meteorology and focus on emissions by running 2018 and 2022 with the same meteorology.

Done, we have now included additional simulations with fixed meteorology in the revised manuscript. The results show that meteorology accounts for about 60–80% of the observed changes in ozone when comparing 2022 to 2018, with emissions playing a smaller but non-negligible role. Looking ahead, continued reductions in precursor emissions may create conditions where even modest meteorological perturbations could trigger disproportionately large increases in ozone or its precursors, which is why understanding the sensitivity of ozone formation regimes to meteorology and emissions are important. We have changed the title and significant parts of the discussion to reflect that the changes in O3 are coming both from meteorology (primarily) and PM10 emissions changes. A detailed discussion is given in **Lines 411–419, 424–426, 439–442, 460–463, 469–472, 476–514. Figures 5 and 6** are modified to include the analysis from the fixed meteorological simulations. Details of the model setup are provided in **lines 188–197**.

Previous work has found that gHO2 of 0.2 might be too high in East Asia https://doi.org/10.5194/acp-23-2465-2023 and references within. How much would a value of 0.1 impact your results? Please include a discussion of the uncertainties surrounding this uptake coefficient. Also see this recent paper on the large variability in gHO2 depending on atmospheric conditions. https://pubs.rsc.org/en/content/articlehtml/2024/ea/d4ea00025k

Done. We have included a new figure, **Figure 4,** showing the difference in ozone levels with and without an aerosol uptake coefficient of 0.1. Compared to the changes observed with a coefficient of 0.2, the ozone difference with 0.1 is approximately half as large. A detailed discussion of these results has been added to the revised MS, **lines 385–402**.

**Specific Comments**

Page 2, line 38:

The author's state: "In the NOx-limited regime, the concentration of NOx is low compared to VOCs, implying that increasing VOC levels can promote further ozone generation due to inadequate NOx to completely react with VOCs." This is incorrect. In the NOx-limited regime, the concentration of NOx is low relative to VOCs, meaning that ozone production is limited by the availability of NOx and increasing VOC levels has little effect. Similarly, the authors state "In a VOC-limited regime, the availability of VOCs is restricted, and further VOC emissions will exert a negligible impact on ozone formation until NOx concentrations are high." This is also incorrect. If ozone chemistry is VOC-limited, adding VOCs will increase ozone production.

Done, we have now corrected this as, "In the NOx-limited regime, the concentration of NOx is low compared to VOCs, implying that *increasing NOx levels* can promote further ozone generation due to inadequate NOx to completely react with VOCs. In a VOC-limited regime, the availability of VOCs is restricted, *and further VOC emissions can significantly enhance ozone formation, whereas additional NOx may have less effect on ozone formation*", in **Lines 37–41**.

Page 2 line 45 – The authors should cite https://www.nature.com/articles/s41561-022-00972-9 and https://www.pnas.org/doi/abs/10.1073/pnas.1812168116 where appropriate in this manuscript.

Done. We have now included these references in the revised MS, **line 45.**

Page 2, line 49 – There must be a typo in 2017–18. since 2019.

Done. We have now removed this typo.

Page 2, line 51 – How many exceedances were there in 2018?

Done. Visakhapatnam and Tirupati experienced about 60 and 10 such days in 2018, which is now mentioned in **line 53**.

Page 3, line 78 – What about the impact of aerosols on modifying PBL dynamics? I believe there is a large body of work also on this topic (e.g., https://academic.oup.com/nsr/article/4/6/810/4191281).

Done, this is now discussed in **Lines 321–325.**

Page 3, line 94 – Cite https://www.nature.com/articles/s41561-022-00972-9 again here.

Done, the reference is included in **Line 98.**

Page 4, line 124 – It is good practice to give the model version number and DOI here.

Done. We use the GEOS Chem version 14.4.3. The version number and doi is now included in **lines 129–130**.

Page 4, line 125 – FlexChem is an option. Do you actually use it? If not, don't cite it.

Done, we have now removed this citation.

Page 4, line 130 - Cite the associated papers, although these advancements don't seem relevant to this work. Generally, just cite the relevant updates that impact your work.

Done, we have now cited associated papers related to redox mechanisms in **Lines 134–135.**

Page 4, paragraph starting on line 138 – It is not necessary to cite GEOS-Chem papers from other regions. Just focus on India, and if necessary other parts of Asia. Other papers to include on India: https://pubs.acs.org/doi/full/10.1021/acsearthspacechem.2c00150. It is most important to show that your modelsetup has good performance against observations.

Done. We have now included this reference in the revised MS, **lines 155–158**.

Page 5, paragraph starting on line 163 – Please provide information on what emissions you have chosen for India, and what scheme you are using to calculate aerosol (simple, complex, etc.). Also please provide information on the observations. Only PM10 is available? How did you calculate PM10 from GEOS-Chem individual species? It looks like there is ozone in the supplement as well? What is the instrument used, and the temporal resolution?

Done. GEOS-Chem uses a set of chemical mechanisms implemented with the Kinetic PreProcessor (KPP) (Damian et al., 2002). The standard chemical mechanism in GEOS-Chem has undergone

continuous development since the original tropospheric gas-phase scheme of Bey et al. (2001), with subsequent extensions to include aerosol chemistry (Park, 2004), stratospheric chemistry (Eastham et al., 2014), and a detailed tropospheric–stratospheric halogen chemistry module (Wang et al., 2019), which include 299 chemical species (Fritz et al., 2022). Sulfate–nitrate–ammonium aerosols are simulated with a bulk thermodynamic approach (Park, 2004), where gas–particle partitioning is calculated using ISORROPIA II (Fountoukis and Nenes, 2007). Sea salt is represented with two modes distinguishing fine and coarse particles (Jaeglé et al., 2011), while mineral dust is described using four particle size bins (Fairlie et al., 2007). Furthermore, the simple SOA scheme is employed for secondary organic aerosols, based on reversible partitioning of semivolatile products of VOC oxidation (Pye et al., 2010). This is now mentioned in the revised MS, **Lines 188–197.**

The heterogeneous radical loss rate of $HO_2$ is determined by iterating through the radius and surface area of various aerosol types, which includes sulfate–nitrate–ammonium thermodynamics, secondary organic aerosol, dust, sea salt, black carbon, and primary organic carbon (Ivatt et al., 2022). These factors, along with temperature and air density, were used to calculate the first-order loss rate of $HO_2$, mentioned in **Lines 257–260.**

The Central Pollution Control Board (CPCB) monitors ambient ozone using UV photometric, chemiluminescence, and chemical methods, while $PM_{10}$ (particles with aerodynamic diameter <10 μm) is measured using gravimetric techniques, Tapered Element Oscillating Microbalance (TEOM), and beta attenuation methods, mentioned in **lines 204–210.**

In GEOS-Chem, $PM_{10}$ is diagnosed as the sum of $PM_{2.5}$ plus size-resolved dust (70% of DST2, all of DST3, 90% of DST4) and coarse sea-salt (SALC scaled by the hygroscopic growth factor, e.g., 1.86 at 35–50% RH), with the final values archived at standard temperature and pressure (STP) using the ideal gas law. This is now mentioned in **lines 198–201.**

Figure S1 – Typically people show observations in black, and the model in red.

Done, we have now revised this figure. Please find the new **Figure S1** in the Supplementary Document.

Page 6, line 180 – If the model is biased that high, there is likely something wrong with emissions. If you are not going to be able to investigate that, then it would be important to show that the model can capture the 2018 vs. 2022 difference to provide support for the relative impacts on ozone from aerosol differences.

Done. We have now included **Figure S4** and **Figure S5** to show that modelled vs observed ozone and PM differences are consistent for the years 2018 and 2022. A corresponding discussion is provided in **Lines 223–234**.

Figure 2 – Could you put the observations as colored circles on top of the model? This would help with the previous comment. It is also important if you are going to discuss drivers of the pollution hotspots to show confidence in the model distribution.

Done. We have now revised **Figure 2** by including the observations as colored circles.

Page 9, line 300 – Be careful of stating too much causation here. In winter there is also low sunlight, and cooler temperatures that promote nitrate formation.

Done, we have now revised this statement in **lines 309–311.**

Page 10, line 305 – BC scavenging of ozone is not in models, how important is it?

Done. Black carbon (BC), a constituent of PM from combustion, can directly scavenge ozone by acting as a deposition surface (Gao et al., 2018). However, this direct BC–ozone scavenging pathway is not explicitly represented in most chemical transport models (Koch et al., 2009), and while its overall contribution is considered modest compared to dominant photochemical production and deposition processes, it may still have localised importance in high-BC environments such as South Asia (Kumar et al., 2015). This is now mentioned in **Lines 315–318**.

Page 10, line 307 – This is not an inverse relationship then if both PM and ozone accumulate near the surface. Also, you haven't mentioned the impact on the PBL height from aerosols (e.g., https://academic.oup.com/nsr/article/4/6/810/4191281). Also, if the PBL is suppressed, you might get ozone titration from high NOx emissions.

Done, In such cases, the co-accumulation of PM and ozone near the surface does not imply a simple inverse relationship, as aerosol–radiation interactions can further suppress the planetary boundary layer (PBL) height (Li et al., 2017; Zou et al., 2017), thereby enhancing pollutant trapping. Under suppressed PBL conditions and high NOx emissions, ozone titration can also occur, leading to additional complexity in the surface ozone response (Lin et al., 2008; Li et al., 2024). This is now mentioned in **Lines 321–325**.

Page 11, line 343 – Does the model have different emissions in 2022 or 2018, or are the same emissions (e.g. for biomass combustion) used for both years?

Yes, the model has different emissions for both years. The emission inventories used in the model are mentioned in **lines 188–197**.

Page 15, line 407 – This is opposite the conclusion of Ivatt et al., 2022, who found that Delhi is VOC-limited. Looking at your map, it looks like Delhi could be either a purple color or red color depending on the season, which would be VOC-limited or aerosol-inhibited.

Done, we have now revised this statement in **lines 413–419**.

Page 15, line 408 – The authors state that "The dominance of peroxyl-radical self-reactions in urban areas such as Delhi, Bhiwadi, Vadodara, Jamshedpur and Nagpur indicates that NOx emissions in these cities limit ozone formation, promoting pathways where peroxy radicals neutralize ozone. This is typical of regions where high NOx levels suppress the production of ozone by inhibiting HOx chemistry (Romer et al., 2018)." This is incorrect! High NOx regions suppress ozone because the dominant loss is NO2 + OH, but this means that these cities are VOC limited.

Done, we have now revised this statement in **lines 413–419**.

On Figure 7 – It looks like Delhi is in the VOC-limited or aerosol-limited regime, depending on season. This contradicts the statement on Line 406 that Delhi (and other urban areas) are NOx-limited.

Done, we have now revised this statement in **lines 413–419**.

Page 17, line 442 – Again there is a need to be careful here as it is unlikely that 100% of the ozone difference is due to the reduction in aerosols. One possible way to remove the impact of meteorology would be to use the same year of meteorology but with different emissions, assuming that different emissions are used in these two years.

Done, we have now included the fixed meteorology simulations in the revised MS, and a detailed discussion is provided in **Lines 411–419, 424–426, 439–442, 460–463, 469–472, 476–514. Figures 5 and 6** are modified to include the analysis from the fixed meteorological simulations.

Page 17, line 454 – Can you give more specific reasons for why the monsoon season would be more susceptible to aerosol uptake than winter or post-monsoon?

Done. *The enhanced sensitivity during the monsoon is attributable to elevated humidity and aerosol liquid water content, stronger photochemical radical production, and deeper boundary layer dynamics, which collectively amplify the influence of aerosol uptake on ozone formation relative to winter or post-monsoon.* This is now mentioned in **lines 476–479**.

Paragraph starting on Page 18, line 480 – How much does reduced NOx also reduce aerosol? What is the composition of aerosol? Is there a lot of ammonium nitrate?

Done. A new figure showing the changes in aerosol surface area due to reduction in primary emissions of NOx is now included as **supplementary Figure 12** and is mentioned in the revised MS, **Lines 534–537.**

**Response to Reviewer #2**
* * *
The study by Gopikrishnan et al. reports the effect of HO2 uptake on aerosols in the years 2018 (high aerosol year) and 2022 (low aerosol year) in the entire India region using the GEOS-Chem model. In 2022, during winter and post-monsoon seasons, HO2 uptake on aerosols reduced ozone while this is reversed during the monsoon season due to lower aerosols. Based on NOx reduction scenarios, 80 % of the increase in ozone due to HO2 uptake on aerosols can be resolved through NOx mitigation. Overall, the study reports important results on the impact of this new chemical regime and the following revisions are suggested for improvement.

Thank You for the comments. Please see the specific response for each comments below:

**Major comments:**
The modeled to observed O3 discrepancies seem significant in Figure S1. The authors provide several factors (lines 176 to 183) that could cause model to obs discrepancies such as uncertainties in the precursor emissions, ozone sinks, and the coarse resolution of the model. However, the modeled O3 is significantly higher than observations in all the ground station reported values shown in Figure S1 throughout the entire year. The model overestimates the obs by up to ~75 ppb (assuming that the y-axis is in µg/m3) and in many periods ~ 50 ppb which seems way too high compared to previous literature on GEOS-Chem model studies in Asia. Since the paper mainly discusses different chemical regimes (NOx- vs VOC- vs aerosol-limited) it seems worthwhile to confirm that the emissions have been set up properly in the model. More details on the model setup should be added in section 2.2. (e.g., meteorology and emission inventory used).

Done, We have now included the model setup in the revised manuscript (**lines 188–197**). Since the focus of this study is on changing chemical regimes, it is essential that the model captures the differences in ozone between 2022 and 2018. The comparison between observations and model results shows good consistency, as presented in **Supplementary Figures S4 and S5,** with further discussion provided in the revised manuscript (**lines 223--234**). However, previous studies also shows that the bias is relatively high compared to CPCB ground-based measurements in a daily scale analysis (Pai et al., 2022; Karambelas et al., 2022)

*References:*
*Pai, S.J., Heald, C.L., Coe, H., Brooks, J., Shephard, M.W., Dammers, E., Apte, J.S., Luo, G., Yu, F., Holmes, C.D. and Venkataraman, C., 2022. Compositional constraints are vital for atmospheric PM2. 5 source attribution over India. ACS earth and space chemistry, 6(10), pp.2432-2445.*
*Karambelas, A., Fiore, A.M., Westervelt, D.M., McNeill, V.F., Randles, C.A., Venkataraman, C., Pierce, J.R., Bilsback, K.R. and Milly, G.P., 2022. Investigating drivers of particulate matter pollution over India and the implications for radiative forcing with GEOS-Chem-TOMAS15. Journal of Geophysical Research: Atmospheres, 127(24), p.e2021JD036195.*

Similar to Ivatt et al. (2022), it would be useful to add sensitivity tests by adjusting the HO2 uptake coefficient as this number has a wide reported range (0.08-0.4) and could change the relative importance of each chemical regime.

Done. We have included a new figure, **Figure S9**, showing the difference in ozone levels with and without an aerosol uptake coefficient of 0.1. Compared to the changes observed with a coefficient of

0.2, the ozone difference with 0.1 is approximately half as large. A detailed discussion of these results has been added to the revised MS, **lines 385–402**.

There are details on the entire region of India that doesn't seem highly relevant to the main point of the paper which makes the paper difficult to read. Throughout sections 3.1.1, 3.1.2, and 3.1.3, the authors describe trends in PM10 or ozone for each region/year/season and provide previous literature studies to derive causations of the modeled trend. In section 4.1 this continues describing a figure in the supporting information. This level of detail can be useful for the interested readers but not necessarily needed for the discussion. I suggest leaving only relevant information in the main text and moving the rest to the supporting information.

Done. We have **shortened section 3** and moved the detailed discussions in Section 3 to **section 1** in the supplementary data. Additionally, we have also moved **section 4.1** to **section 2** in the supplementary data.

Minor comments:

(lines 37 − 41) The statements about NOx-limited and VOC-limited seem reversed. In a NOx-limited regime, radical-radical termination reactions dominate, and higher levels of VOCs won't lead to significant O3 increase. In a VOC-limited environment, loss through NO2 + OH dominates and reducing VOCs will be more efficient in lowering O3. Also, NOx "concentrations" are not directly compared to VOC "concentrations" so I would revise "concentration of NOx is low compared to VOCs".

Done, we have now corrected this in the revised MS, **Lines 37--41**.

Line 49 Add definition for "MDA-8"

Done, the definition is now mentioned in **Lines 50--51**.

Line 51 How many days of ozone exceedances were reported for the 2018 base year?

Done. Visakhapatnam and Tirupati experienced about 60 and 10 such days in 2018, which is now mentioned in **lines 53**.

Line 51 Should 100 ppb be 100 µg/m3?

Done, The mentioned study uses a limit of 100 ppb.

Figure 1 Colors for NW and Central are too similar. Suggest changing the color of one of them. Northeast India is shown as "NE" in the figure legend but "NEI" in the caption.

Done, We have now consistently used "NE" for North East India in the MS.

Figure S1 Suggest adding units on the y-axis.

Done, we have now included the unit in the y-axis in the revised MS, **Figure S3**.

Line 300 The relationship between PM10 and ozone is not always inverse. Should clarify.

Done, this relation is often inverse, however, also varies substantially depending on meteorological conditions and precursor availability. This is now included in the revised MS, **Lines 309--311**

Line 354 western -> Western

Done, this is now in the supplementary section of the revised MS, **Line 125.**

Figure 6 The color scale is a little difficult to see, especially the blue and red gradations. Suggest adjusting.

Done. We have adjusted the figure to enhance the visibility of the color scale. As the figure uses an RGB triangular colorbar, the scope for further adjustment is inherently limited. We hope the revised version is now clearer and acceptable.

Lines 406-410 If the peroxy-radical self-reactions dominate, it is the VOCs that suppress ozone through the self-reactions.

Done, we have now corrected this statement in the revised MS, **Lines 411–419**.

Line 512 "Furthermore, scaling down the NOx emissions in the model simulations by 25 and 50 % shows a reduction of about 5-10 and 10-15 µg/m3 of surface ozone in India" Are these numbers referring to the bottom graph in Figure 8? It seems more like 1-4 µg/m3 for a 25 % and 4-8 µg/m3 for a 50 % reduction.

Done. Scaling down the NOx emissions in the model simulations by 25 and 50% shows a *spatial variability* in the reduction of surface ozone, with decreases of about 5–10 and 10–15 µg/m³ across India, respectively. We have now clarified this in **line 575–577**.

V03/21092025